# Effects of Pyrrole-Imidazole Polyamides Targeting Human TGF-β1 on the Malignant Phenotypes of Liver Cancer Cells

**DOI:** 10.3390/molecules25122883

**Published:** 2020-06-23

**Authors:** Keiko Takagi, Yutaka Midorikawa, Tadatoshi Takayama, Hayato Abe, Kyoko Fujiwara, Masayoshi Soma, Hiroki Nagase, Toshio Miki, Noboru Fukuda

**Affiliations:** 1Department of Digestive Surgery, Nihon University School of Medicine, Tokyo 173-8610, Japan; takagi.keiko55@nihon-u.ac.jp (K.T.); takayama.tadatoshi@nihon-u.ac.jp (T.T.); abe.hayato@nihon-u.ac.jp (H.A.); 2Division of General Medicine, Department of Medicine, Nihon University School of Medicine, Tokyo 101-8309, Japan; fujiwara.kyoko@nihon-u.ac.jp (K.F.); soma.masayoshi@nihon-u.ac.jp (M.S.); 3Department of Anatomy, Nihon University School of Dentistry, Tokyo 101-8310, Japan; 4Sasaki Foundation, Department of Medicine, Kyoundo Hospital, Tokyo 101-0062, Japan; 5Laboratory of Cancer Genetics, Chiba Cancer Center Research Institute, Chiba 260-8717, Japan; hnagase@chiba-cc.jp; 6Department of Physiology, Nihon University School of Medicine, Tokyo 173-8610, Japan; miki.toshio@nihon-u.ac.jp; 7Division of Nephrology, Hypertension and Endocrinology, Department of Medicine, Nihon University School of Medicine, Tokyo 173-8610, Japan; fukuda.noboru@nihon-u.ac.jp

**Keywords:** pyrrole-imidazole polyamide, TGF-β1, liver cancer, cancer stem cell, novel candidate drug

## Abstract

Synthetic pyrrole-imidazole (PI) polyamides bind to the minor groove of double-helical DNA with high affinity and specificity, and inhibit the transcription of corresponding genes. In liver cancer, transforming growth factor (TGF)-β expression is correlated with tumor grade, and high-grade liver cancer tissues express epithelial-mesenchymal transition markers. TGF-β1 was reported to be involved in cancer development by transforming precancer cells to cancer stem cells (CSCs). This study aimed to evaluate the effects of TGF-β1-targeting PI polyamide on the growth of liver cancer cells and CSCs and their TGF-β1 expression. We analyzed TGF-β1 expression level after the administration of GB1101, a PI polyamide that targets human TGF-β1 promoter, and examined its effects on cell proliferation, invasiveness, and TGF-β1 mRNA expression level. GB1101 treatment dose-dependently decreased TGF-β1 mRNA levels in HepG2 and HLF cells, and inhibited HepG2 colony formation associated with downregulation of TGF-β1 mRNA. Although GB1101 did not substantially inhibit the proliferation of HepG2 cells compared to untreated control cells, GB1101 significantly suppressed the invasion of HLF cells, which displayed high expression of CD44, a marker for CSCs. Furthermore, GB1101 significantly inhibited HLF cell sphere formation by inhibiting TGF-β1 expression, in addition to suppressing the proliferation of HLE and HLF cells. Taken together, GB1101 reduced TGF-β1 expression in liver cancer cells and suppressed cell invasion; therefore, GB1101 is a novel candidate drug for the treatment of liver cancer.

## 1. Introduction

Transforming growth factor (TGF)-β is a cytokine that regulates the differentiation and growth of transforming cells. In liver cancer, TGF-β accelerates invasion [1], metastasis [2], and angiogenesis of the tumors [3]. TGF-β is also responsible for the induction of epithelial-mesenchymal transition (EMT), which enables tumor invasion and the growth of liver cancer tissue [4]. Inhibition of TGF-β is known to suppress TGF-β-mediated infiltration of liver cancer cells into blood vessels and surrounding stroma [5]. Thus, suppressing TGF-β function is thought to inhibit the progression of liver cancer; nevertheless, there are currently no effective drugs available to suppress TGF-β function for clinical use.

Cancer stem cells (CSCs) are pivotal for cancer maintenance [6] and TGF-β1 is known to be involved in cancer development by transforming precancer cells to CSCs [7]. TGF-β1 was also reported to promote the invasive property of CD44-positive hepatocellular carcinoma (HCC) by inducing mesenchymal transformation [8].

To explore a new strategy to inhibit TGF-β in liver cancer, we evaluated the effect of pyrrole-imidazole (PI) polyamide targeting TGF-β in liver cancer cells and CSCs. PI polyamides are novel gene silencers that were initially identified from antibiotics such as distamycin A and duocarmycin A [9]. Small synthetic PI polyamide molecules are composed of the aromatic amino acids N-methylpyrrole and N-methylimidazole that recognize and bind DNA with sequence specificity [10]. Due to their ability to bind double-helical DNA with high affinity and specificity, PI polyamides can inhibit DNA interaction with proteins including transcription factors [11]. Previous reports demonstrated that PI polyamide targeting human TGF-β1 promoter could significantly inhibit both TGF-β1 promoter activity and the expression of TGF-β1 mRNA and protein in human cells [12,13]. PI polyamide targeting human TGF-β1 was also shown to inhibit the EMT of human fibroblast and promote the induction of induced pluripotent stem cells [14].

We previously reported that seven PI polyamides (GB1101-1107) were examined for the suppression of TGF-β1 mRNA which was stimulated by phorbol 12-myristate 13-acetate, and GB1101 and 1106 significantly inhibited expression of TGF-β1 in marmoset-derived fibroblasts [15]. Given that the former agent could strongly suppress the TGF-β1 expression in a dose-dependent manner, we determined GB1101 as a lead compound of PI polyamides to TGF-β1. In this study, we assessed the effect of GB1101 targeting human TGF-β1 on the growth of liver cancer cells and CSCs and their *TGF-β1* expression levels.

This study aimed to evaluate the effects of TGF-β1-targeting PI polyamide (GB1101) on the growth of liver cancer cells and CSCs and their TGF-β1 expression.

## 2. Results

### 2.1. Effects of TGF-β1-Specific PI Polyamide in HepG2 Cells

To investigate whether GB1101 could suppress TGF-β1 transcription, we evaluated expression TGF-β1 mRNA in HepG2 and HLF cells with or without GB1101 treatment by qPCR. GB1101 treatment dose-dependently decreased abundance of TGF-β1 mRNA, with 1μM GB1101 significantly reduced TGF-β1 mRNA level in HepG2 (*p* < 0.05) and HLF cells (*p* < 0.01) when compared to untreated control cells (Figure 1a,b). In contrast, mismatch PI polyamide at 0.5 to 5 μM did not reduce the abundance of TGF-β1 mRNA in HepG2 cells when compared to the control (Figure 1c).

### 2.2. Effects of PI Polyamide on the Growth of Liver Cancer Cells

We subsequently evaluated whether GB1101 treatment can also repress the growth of liver cancer cells. In a colony formation assay of HepG2 cells, the number of colonies with over 100 μm in diameter were counted on day 14 post-GB1101 treatment (Figure 2). The average number of colonies per field of view was 8.3 ± 2.3 in the control, and 9.6 ± 1.5, 4.0 ± 1.0, 4.3 ± 0.6, and 4.3 ± 0.3 following 0.5, 1, 2, 3 μM polyamide treatment, respectively. Colony formation appeared to be suppressed at polyamide concentrations of 1 to 3 μM, although the reductions were not statistically significant. These results indicated that GB1101 could suppress the growth of HepG2 cells by inhibiting TGF-β1 transcription.

### 2.3. Effects of TGF-β1 PI Polyamide on the Invasion Capacity of HepG2 and HLF Cells

We firstly evaluated *CD44* expression as a marker of CSCs in four liver cancer cell lines (HLE, HLF, Huh7, and HepG2). HLE and HLF cells highly expressed *CD44* mRNA (Figure 3a), while Huh7 and HepG2 cells showed lower CD44 mRNA expression. We then examined effects of TGF-β1 PI polyamide on the invasion capacity of HepG2 and HLF cells with low and high CD44 expression, respectively. The results showed that GB1101 did not appreciably inhibit the invasion of HepG2 cells when compared to untreated cells (*p* = 0.54) (Figure 3b). In contrast, treatment with 3 μM GB1101 significantly (*p* < 0.01) suppressed the invasion of HLF cells (Figure 3c), suggesting that GB1101 specifically inhibits CSCs formation with the suppression of TGF-β1.

### 2.4. Effects of TGF-β1 PI Polyamide on Sphere Formation of CSCs

To further determine the effects of TGF-β1-targeting PI polyamide to CSCs, we performed a sphere formation assay of HLF and HLE cells. We confirmed that treatment with 1 μM GB1101 results of the sphere formation assay showed that 3 ± 3 spheres and 5 ± 2 spheres in HLF cells (*p* = 0.27) and 9 ± 3 spheres and 16 ± 10 spheres were observed in the presence or absence of GB1101 in HLE cells (*p* = 0.19) (Figure 4a,b). There was no statistically significant difference, but the ability to form a sphere tended to be suppressed by GB1101 administration. Furthermore, GB1101 significantly (*p* < 0.05) suppressed abundances of TGF-β1 mRNA in HLE and HLF sphere-forming cells, respectively (Figure 4c), suggesting that GB1101 can suppress the growth of liver cancer cells by inhibiting liver CSCs.

## 3. Discussion

Results of this study showed that TGF-β1 expression was significantly suppressed in HepG2 cells treated with GB1101. In a matrigel invasion assay of HLF cells, we also demonstrated that GB1101 suppressed the invasion of CSC. Furthermore, GB1101 suppressed expression of TGF-β1 mRNA expression and sphere formation of HLE and HLF cells. Thus, our results indicated that GB1101 could suppress cell growth and invasion of liver cancer cells through TGF-β1 downregulation.

TGF-β is a cytokine that is required for the anchorage-independent growth of fibroblasts. TGF-β plays a growth inhibitory role in the early stage of cancer, but shows proliferation-promoting effects at advanced stages of cancer. In this study, we examined the effects of GB1101, a PI polyamide that targets human TGF-β1, on TGF-β1 expression and growth of HCC. PI polyamides bind to the double-helical DNA minor groove with high affinity and specificity, and competitively inhibit binding of transcription factors to the promoter region of target genes. PI polyamides can be taken up by cells and transported to cell nuclei without any special drug delivery systems; thus, PI polyamides are considered to be stable rather than nucleic acid-based drugs such as antisense DNA and siRNA, rendering them as promising new molecules for gene silencing in vivo [16]. Furthermore, we previously performed microarray analysis in renal cortex from salt-sensitive hypertensive rats treated with TGF-β1 targeting PI polyamide. We found that decreased transcripts were almost TGF-β1-related genes. These findings indicate that even PI polyamides bind to many genes, however, they specifically suppressed the transcriptionally activated genes in the disease state [17].

We have previously demonstrated that GB1101 binds adjacent to the FSE2 site in human TGF-β1 promoter and effectively inhibits the expression of phorbol 12-myristate 13-acetate-stimulated TGF-β1 transcription [15]. PMA is known to stimulate TGF-β1 promoter activity via the AP-1 site, but not through the FSE2 site [18]. Nevertheless, the binding of PI polyamide was reported to alter the conformation of double-stranded DNA promoter construction and impair target promoter activity [19]. We also reported that PI polyamides targeting the *TGF-β1* promoter improved TGF-β1-induced fibrotic diseases such as progressive renal diseases [20], diabetic nephropathy [21], hypertrophic scar in marmoset [15], restenosis of arteries after injury [22], encapsulating peritoneal sclerosis [23], and suppressed dimethylnitrosamine-induced liver fibrosis [24]. In addition, pyridine derivative that inhibits TGF-β production improve HCV-related fibrosis and inflammation [25]. Therefore, GB1101 administered for liver cancer patients may improve the coexisting liver fibrosis. Given that a cell-viability assay on cultured hepatocyte cell lines showed that PI polyamide treatments induced no significant reduction in viability or increase in cell death [26], there was no effect on cytotoxicity of PI polyamide on primary cells such as lymphocytes.

On the other hand, due to the multiple anti-inflammatory activities, TGF-β1 is thought to inhibit several aspects of the immune response, and this cytokine could have an immune-down-regulatory role in the pathophysiology of chronic progressive multiple sclerosis [27]. Furthermore, it has been reported that TGF-β1 blood level of patients with major depressive disorder was lower compared to healthy controls [28]. Therefore, GB1101 has possibility to exacerbate the symptoms of such patients as side effects.

The SMAD proteins, intracellular effector of TGF-β signalings, are activated by type I and II TGF-β receptors and translocates into the nucleus to regulate their transcriptions [29]. The TGF-β type I receptor inhibitor known as the activin like kinase 5 inhibitor, is a cell membrane permeable, selective ATP antagonist that was previously shown to inhibit TGF-β activity in HepG2 cells [30], a cell line derived from human liver cancer that have been extensively used as an alternative model of the liver in a wide range of studies, such as carcinogenesis, toxicology, molecular biology, and in screening of anticancer drug candidates [31]. Therefore, a polyamide that targets TGF-β1 is expected to suppress the malignant phenotype of liver cancer cells.

Liver CSCs were firstly reported as a side population cell fraction in a flow cytometry [32], and side population cell sorting has been widely employed as a method to isolate CSCs. CSCs are the source of all malignant cells of primary tumors that are metastatic and resistant to anticancer drugs, and known to be associated with high tumor recurrence rate after treatment [33]. Subsequent studies showed that liver CSCs express markers such as CD133 [34], CD90 [35], CD44 [36,37], and EpCAM [38,39], and possess self-renewing ability and tumorigenicity. Yamashita et al. reported that during the development of liver cancer, CD90-positive CSCs regulate distant organ metastasis and tumorigenicity through the activation of TGF-β signaling when the cells co-exist with EpCAM-positive cells, but not when EpCAM-positive cells exist alone [40]. Other studies have also found that HLE and HLF cells are CD44-positive [8,41].

In the present study, we confirmed that both HLE and HLF cells highly expressed CD44, a marker of CSC. A previous report demonstrated that E-cadherin was upregulated and vimentin was downregulated in HLE and HLF cells, and CD44-positive HCC was shown to be invasive, leading to poor prognosis [42]. These findings suggested that CD44-positive HCC is regulated by CSC and invasive through the EMT with TGF-β1. Thus, we hypothesized that the PI polyamide targeting human TGF-β1 can suppress the development of HCC by inhibiting TGF-β1 transcriptional activity and examined its effects on the invasion and sphere formation of liver CSCs.

There are several limitations in this study. First, we did not confirm inhibition of cancer progression via TGF-β1 suppression by GB1101 in protein level. However, we previously reported the GB1101 effects on various types of diseases such as nephropathy [20], hypertrophic scar [15], restenosis of arteries after injury [22], peritoneal sclerosis [23], and liver fibrosis [24] using rat or marmoset models. Therefore, the perspective of this study is to evaluate whether GB1101 would actually be available to cancer therapy by in vivo study. Second, given that polypharmacy using anti-cancer drugs and molecular drugs is a mainstream for cancer treatment, synergistic or antagonistic potential interference with GB1101 should have been investigated. Then we are now planning to identify the drugs that enhance the GB1101 ability to kill the cancer cells.

In this study, we performed a sphere formation assay to quantify CSCs by counting the number of spheres formed in the culture system. Cells within the spheres possess a capacity for self-renewal and tumorigenesis, and the sphering of HLE and HLF cells are considered equivalent to hepatic CSCs [43]. In our study, we found that the TGF-β1-inhibitory polyamide suppressed *TGF-β1* expression in sphere-forming HLE and HLF cells.

## 4. Materials and Methods

### 4.1. Design and Synthesis of PI Polyamides Targeting the hTGF-β1 Promoter

We previously designed seven PI polyamides that bind to human *TGF-β1* promoter [21]. In this study, we evaluated the effect of one of the PI polyamides, GB1101, that targets sequences adjacent to the FSE2 regulatory element in the hTGF-β1 promoter. The structure of GB1101 and a mismatch PI polyamide is shown in Appendix A [21]. Machine-assisted automatic synthesis of hairpin-type PI polyamides was carried out using a continuous-flow peptide synthesizer (PSSM-8; Shimadzu, Kyoto, Japan) at 0.1 mmol scale (200 mg of Fmoc-b-alanine CLEAR Acid Resin at 0.50 meq/g; Peptide Institute, Osaka, Japan). The automatic solid phase synthesis was performed as described previously [15].

### 4.2. Cell Culture

The hepatoblastoma cell line HepG2 and HCC cell line Huh7 were obtained from RIKEN (Tokyo, Japan). HCC cell lines HLE and HLF were purchased from Japanese Collection of Research Biosources (Osaka, Japan). HepG2, HLE, and HLF were maintained in DMEM (Sigma-Aldrich, Tokyo, Japan) medium and Huh7 was maintained in PRMI1640 (Sigma-Aldrich, Japan) medium supplemented with 10% (HLF only 5%) fetal calf serum (Invitrogen, Carlsbad, CA, USA) and 50 mg/mL streptomycin (Invitrogen) at 37 °C under 5% CO_2_ in air.

### 4.3. Assessment of TGF-β1 and CD44 mRNA Expression

For the evaluation of *TGF-β1* expression, HepG2 or HLF cells were cultured for 24 h and then incubated with 0.01, 0.1, or 1.0 μM GB1101 in DMEM for 24 h. Subsequently, total RNA was isolated and reverse-transcribed to cDNA as described previously [44]. Real-time quantitative PCR (qPCR) was performed using the cDNA and SYBR Premix EX TaqII (Takara Bio Inc, Shiga, Japan) in Thermal Cycler Dice Real Time System TP800 (Takara Bio Inc., Shiga, Japan). Primers used for the qPCR were: TGF-β forward (5′-CTGGACACCAACTATTGC-3′) and reverse (5′-CTTCCAGCCGAGGTCCTT-3′); GAPDH forward (5′-ACCTGACCTGCCGTCTAGAA-3′) and reverse (5′-TGGTGAAGACGCCAGTGGA-3′). The PCR conditions were: one cycle at 95 °C for 30 s, followed by 40 cycles at 95 °C for 5 s and 60 °C for 30 s. The TGF-β expression was normalized to GAPDH expression, and each experiment was performed in triplicate.

For the *CD44* expression analysis, HepG2, Huh7, HLE, and HLF cells were cultured for 24 h and harvested for total RNA solation and reverse-transcription to cDNA. Real-time qPCR was performed as described above, using CD44-specific primers: forward (5′-gcagtcaacagtcgaagaagg-3′) and reverse (5′-tgtcctccacagctccatt-3′). The PCR conditions were: one cycle at 95 °C for 30 s, followed by 45 cycles of denaturation at 95 °C for 5 s, annealing at 58 °C for 10 s, and extension at 72 °C for 20 s.

### 4.4. Colony-Formation Assay

For the colony-formation assays, 2 × 10^5^ cells/mL of HepG2 cell suspension was prepared and cultured in DMEM supplemented with 10% fetal calf serum. Soft agar (5%; Beyotime Institute of Biotechnology, Jiangsu, China) was mixed thoroughly with the medium at a ratio of 1:9, added to the plates, and set aside at room temperature to allow agar to solidify. Subsequently, 1.5 mL of the cell suspension was mixed with an equal volume of 0.5% soft agar, and seeded onto the plates with soft agar. Twenty-four hours after seeding, 0.5 to 3 μM of GB1101 were added to the cells. The plates were incubated at 37 °C with 5% CO_2_ for two weeks, and microscopic colonies > 100 μm in diameter were counted in each well. The experiments were performed in triplicate.

### 4.5. Cell Invasion Assay

The invasive capacity of HepG2 and HLF cells was determined using a transwell system (6.4 mm transwell with 8.0 µm pore polycarbonate membrane insert) coated with extracellular matrix gel (BD BioCoat; Corning Inc., Corning, NY, USA). HepG2 or HLF cells (5 × 10^5^ cells/well) were suspended in MEM or DMEM and seeded into the Matrigel-coated upper chamber and incubated in the presence of 10% fetal calf serum. HepG2- or HLF-conditioned culture supernatant was placed in the lower chamber (6-well plate). The HepG2 cells were incubated for 24 h with 2 μM GB1101, and HLF cells were incubated for 9 h with 3 μM GB1101, and cells invading to the lower chamber were subsequently fixed with 4% paraformaldehyde, stained with 0.1% crystal violet, and counted.

### 4.6. Induction of HLF and HLE Sphere Formation

HLF and HLE cells (1.1 × 10^5^ cells/mL) were suspended and grown in Hanks’ balanced salt solution medium (Sigma-Aldrich, St. Louis, MO, USA) supplemented with 2% fetal calf serum, 20 ng/mL EGF (PeproTech, Rocky Hill, NJ, USA), 20 ng/mL FGF (PeproTech), and 10 mM HEPES in ultralow attachment 6-well plates (Corning Inc.). The cells were incubated with GB1101, and the medium was changed every other day. On day 7, cell spheres > 100 mm in diameter were counted and photographed, and *TGF-β1* expression level in the sphere-forming HLE and HLF cells was analyzed as described above.

### 4.7. Statistical Analysis

Statistical analyses were performed using the JMP 10 software (SAS, Tokyo, Japan), and statistical significance was calculated using student’s *t*-test. All cell culture experiments were performed in triplicate, and the values of continuous variables were presented as the mean ± standard deviation. *P* values of <0.05 were considered statistically significant.

## 5. Conclusions

GB1101, a PI polyamide that targets the human *TGF-β1* promoter, suppressed *TGF-β1* expression and may act to inhibit the anchorage-independent proliferation and invasion of liver cancer cells. In addition, GB1101 inhibited the formation of cell spheres and TGF-β expression in the sphere cells. Together, these results suggested the potential of GB1101 as a novel anticancer drug for liver cancer.

## Figures and Tables

**Figure 1 molecules-25-02883-f001:**
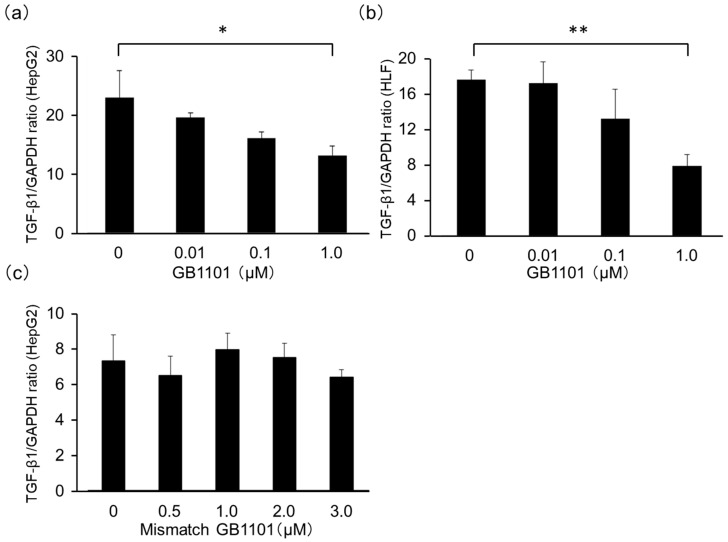
Effect of GB1101 and a mismatch pyrrole-imidazole (PI) polyamide on *TGF-β1* mRNA expression in liver cancer cells. Human hepatoblastoma cell line HepG2 and HLF were cultured in the absence or presence of 0.01 to 1μM GB1101 (**a**,**b**) or 0.5 to 5 μM mismatch polyamide (**c**) for 48 h. Total RNA was extracted to assess the level of transforming growth factor (*TGF)-β1* mRNA by real-time qPCR. Data are presented as the mean ± SD (n = 3). * *p* < 0.05, ** *p* < 0.01.

**Figure 2 molecules-25-02883-f002:**
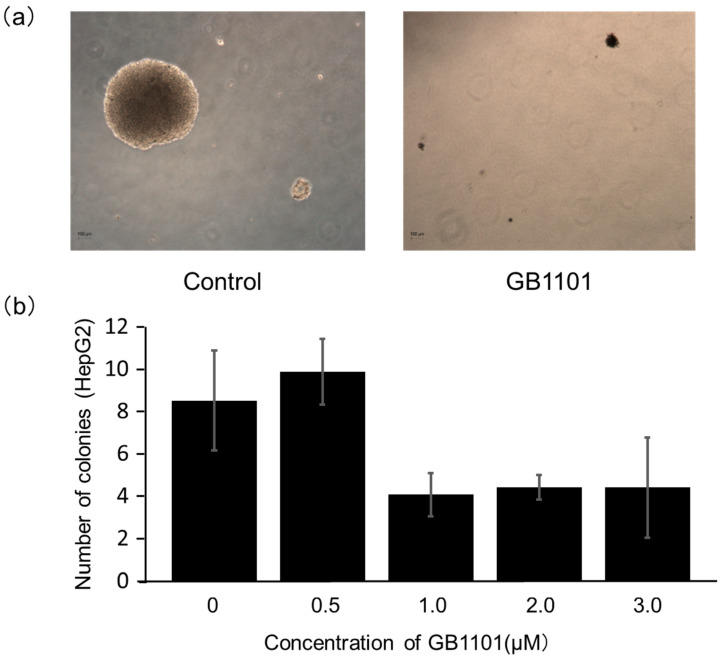
Colony formation of HepG2 cells treated with GB1101. (**a**) Human hepatoblastoma HepG2 cells were cultured in soft agarose in the absence or presence of 0.5 to 3 μM GB1101. (**b**) Colonies were counted 14 days after treatment. Data are presented as the mean ± SD (n = 3).

**Figure 3 molecules-25-02883-f003:**
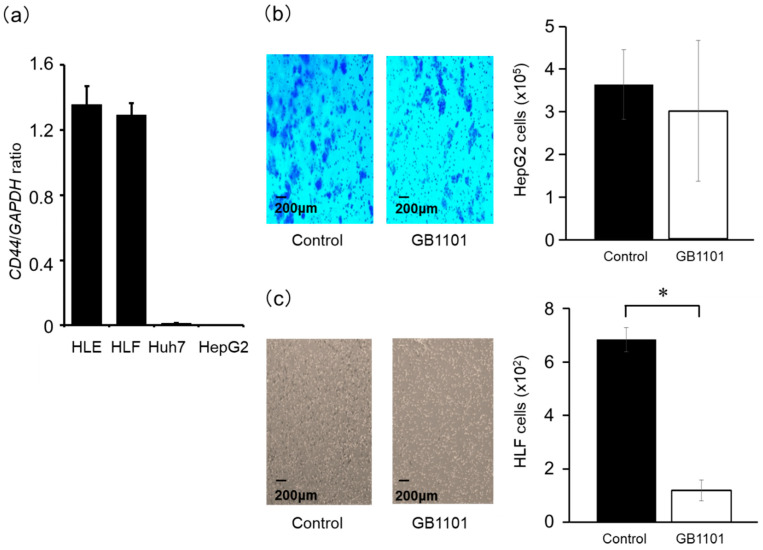
*CD44* expression of hepatocellular carcinoma cells, and effect of GB1101 on the invasion capacity of hepatocellular carcinoma cells in matrigel invasion assay. (**a**) The relative expression level of *CD44* mRNA in four liver cancer cell lines (HLE, HLF, Huh7, and HepG2) was assessed by qPCR. HepG2 and HLF cells were seeded in each invasion chamber at a density of 5 × 10^5^ cells/mL. (**b**) HepG2 cells were treated with 2 μM GB1101 for 24 h, and (**c**) HLF cells were treated with 3 μM GB1101 for 9 h. Representative images of invasive cells are shown, and the number of invasive cells are counted and presented as the mean ± SD (n = 3). * *p* < 0.01.

**Figure 4 molecules-25-02883-f004:**
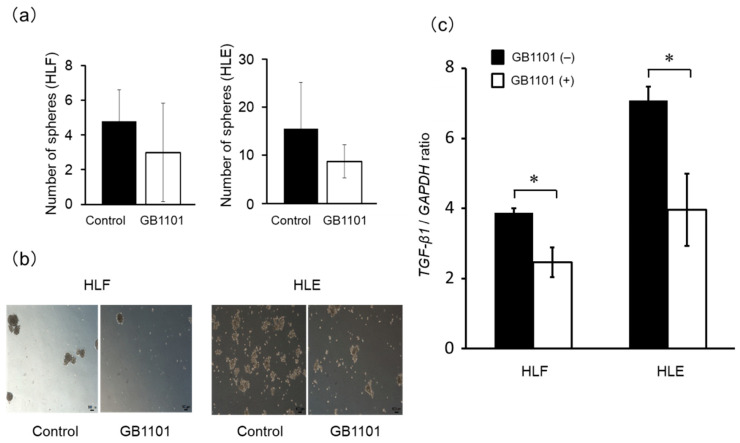
Effect of GB1101 on sphere formation of human HLF and HLE cells, TGF-β1 expression of liver cancer cells. HLF and HLE cells in suspension were incubated in the absence or presence of 1 μM GB1101 for 7 days. The number of spheres formed in the suspension culture medium was counted and the results are presented as the mean ± SD (n = 5) (**a**). At 7 days post-treatment, the cells were photographed (**b**) and TGF-β1 mRNA expression (**c**) was assessed by qPCR and the data are presented as the mean ± SD (n = 3). * *p* < 0.05.

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
