# Peer review of "Effects of Pyrrole-Imidazole Polyamides Targeting Human TGF-β1 on the Malignant Phenotypes of Liver Cancer Cells"

_molecules, 2020, doi:10.3390/molecules25122883_

Round 1

Reviewer 1 Report

In this manuscript, the authors study the effects of a Pyrrole-Imidazole (PI) polyamide on the development of liver cancer cells and their ability to become malignant. The authors base the rationale of their study on former results obtained with such DNA-binding polyamides on a large variety of cancers. A series of seven derivatives has been previously designed to target a human TGF-b1 promoter and the properties of one of them, GB1101, is evaluated for its effects on TGF-b1 expression, liver cells proliferation and invasiveness.

In all the manuscript, the authors only study one PI derivative, that is GB1101. If some former data and reports on the use and biological properties of effects of PI polyamides are well reported in the Introduction, the choice of one of them, that is GB1101, should be better explained. Some information on this choice can be found in other several parts of the manuscript (such as parts 3 and 4.1), but some more explanations should be given as soon as in the Introduction to better understand the experiments presented in part 2 Results.  It is not also clear if the authors have evaluated the specificity of GB1101 on the human TGF-b1 promoter. Such specificity is essential to assess correlations between TGF-b1 expression inhibition and anticancer activities.

In all Figures, “PI polyamide” (and PIP in Figure 4c) should be replaced by the effective PI derivative tested, GB1101.

In the Introduction, lines 62 to 69 must be removed since they have not to appear in the manuscript.

In the Discussion part (line 148), the authors report results from one of their former studies on GB1101 binding to TGF-b1 promoter but the reference [16] is not correct since none of the authors participated to such reference paper.

For more clarity in the discussion process, it might be suggested to change the order of two paragraphs, that is to move lines 170- 179 before line 163.

The authors have to choice for the name of their PI derivative (GB1101 or GB 1101) with (line 120) or without space. Reference 15 (indicated in S1) should also be mentioned line 190.

Author Response

Reviewer 1

In this manuscript, the authors study the effects of a Pyrrole-Imidazole (PI) polyamide on the development of liver cancer cells and their ability to become malignant. The authors base the rationale of their study on former results obtained with such DNA-binding polyamides on a large variety of cancers. A series of seven derivatives has been previously designed to target a human TGF-b1 promoter and the properties of one of them, GB1101, is evaluated for its effects on TGF-b1 expression, liver cells proliferation and invasiveness.

In all the manuscript, the authors only study one PI derivative, that is GB1101. If some former data and reports on the use and biological properties of effects of PI polyamides are well reported in the Introduction, the choice of one of them, that is GB1101, should be better explained. Some information on this choice can be found in other several parts of the manuscript (such as parts 3 and 4.1), but some more explanations should be given as soon as in the Introduction to better understand the experiments presented in part 2

We previously reported that seven PI derivatives (GB1101-1107) were examined for suppression of TGF-β1 expression which was stimulated by phorbol 12-myristate 13-acetate, and found that GB1101 could inhibit it most effectively and dose-dependently (ref 15). Therefore, we applied GB1101 for all the experiments in this study. As indicated by the reviewer, we clearly described it in Introduction.

“We previously reported that seven PI polyamides (GB1101-1107) were examined for the suppression of TGF-β1 mRNA which was stimulated by phorbol 12-myristate 13-acetate, and GB1101 and 1106 significantly inhibited expression of TGF-β1 in marmoset-derived fibroblasts [15]. Given that the former agent could strongly suppress the TGF-β1 expression by dose-dependent manner, we determined GB1101 as a lead compound of PI polyamides to TGF-β1. In this study, we assessed the effect of GB1101 targeting human TGF-β1 on the growth of liver cancer cells and CSCs and their TGF-β1 expression levels.”

(line 61-67)

Results. It is not also clear if the authors have evaluated the specificity of GB1101 on the human TGF-b1 promoter. Such specificity is essential to assess correlations between TGF-b1 expression inhibition and anticancer activities.

As indicated by the reviewer, we clearly described in Discussion that we previously evaluated the binding sites in TGF-β1 promoter region by seven PI polyamides and found that GB1101 binds adjacent to FSE2 site [ref 15].

“We have previously demonstrated that GB1101 binds adjacent to the FSE2 site in human TGF-β1 promoter and effectively inhibits the expression of phorbol 12-myristate 13-acetate-stimulated TGF-β1 transcription [15]. PMA is known to stimulate TGF-β1 promoter activity via the AP-1 site, but not through the FSE2 site [18]. Nevertheless, the binding of PI polyamide was reported to alter the conformation of double-stranded DNA promoter construction and impair target promoter activity [19].”

(line 149-154)

In all Figures, “PI polyamide” (and PIP in Figure 4c) should be replaced by the effective PI derivative tested, GB1101.

As indicated by the reviewer, “PI polyamide” and “PIP” were replaced by “GB1101” in all figures.

In the Introduction, lines 62 to 69 must be removed since they have not to appear in the manuscript.

As indicated by the reviewer, lines 62 to 69 were removed.

In the Discussion part (line 148), the authors report results from one of their former studies on GB1101 binding to TGF-b1 promoter but the reference [16] is not correct since none of the authors participated to such reference paper.

For more clarity in the discussion process, it might be suggested to change the order of two paragraphs, that is to move lines 170- 179 before line 163.

As indicated by the reviewer, the reference was corrected, which was one of our works using GB1101.

(line 335–338)

Also, we changed the order of the two paragraphs as indicated.

(line 177–186)

The authors have to choice for the name of their PI derivative (GB1101 or GB 1101) with (line 120) or without space. Reference 15 (indicated in S1) should also be mentioned line 190.

As indicated by the reviewer, the drug name of PI derivative used in this study was unified into GB1101 (without space).

(line 119)

Reviewer 2 Report

  1. There are several major problem in introduction, which shown non-related paragraph in introduction from line 62 to 69. 
  2. Authors used CD44 as cancer stem cell marker. However, authors only shown the mRNA rather then protein levels of CD44, not convincing. Similar problem also shown in protein levels of TGF-beta.
  3. Since the polyamides bind to minor grooves of DNA, it may show other cellular function in other genes, specificity is a important issue, authors need to verify it.

Author Response

Reviewer 2

  1. There are several major problem in introduction, which shown non-related paragraph in introduction from line 62 to 69.

As indicated by the reviewer, lines 62 to 69 were removed.

  1. Authors used CD44 as cancer stem cell marker. However, authors only shown the mRNA rather then protein levels of CD44, not convincing. Similar problem also shown in protein levels of TGF-beta.

As indicated by the reviewer, it is necessary to validate the GB1101 effects on cancer in protein level. We’ve not done yet but we previously reported the GB1101 effects on several types of diseases using rat or marmoset models. Therefore, we referred to an in vivo study for cancer treatment by GB1101 as one of perspectives in Discussion.

“There are several limitations in this study. First, we did not confirm inhibition of cancer progression via TGF-β1 suppression by GB1101 in protein level. However, we previously reported the GB1101 effects on various types of diseases such as nephropathy [20], hypertrophic scar [15], restenosis of arteries after injury [22], peritoneal sclerosis [23], and liver fibrosis [24] using rat or marmoset models. Therefore, the perspectives of this study is to evaluate whether GB1101 would actually be available to cancer therapy by in vivo study.”

(line 194−199)

  1. Since the polyamides bind to minor grooves of DNA, it may show other cellular function in other genes, specificity is a important issue, authors need to verify it.

Regarding the specificity of PI polyamides to targeting genes, we previously performed microarray analysis in renal cortex from salt-sensitive hypertensive rats treated with TGF-b1 targeting PI polyamide. In this study, decreased transcripts were almost TGF-b1-related genes. Therefore we assumed that even PI polyamides bind many genes, however, they specifically suppressed the transcriptionally activated genes in the disease state.

“Furthermore, we previously performed microarray analysis in renal cortex from salt-sensitive hypertensive rats treated with TGF-β1 targeting PI polyamide. We found that decreased transcripts were almost TGF-β1-related genes. These findings indicate that even PI polyamides bind to many genes, however, they specifically suppressed the transcriptionally activated genes in the disease state [17].”

(line 144−148)

Reviewer 3 Report

This in vitro study demonstrates that synthetic pyrrole-imidazole (PI) polyamide GB1101  dose-dependently decreased TGF-β1 mRNA levels in HepG2 and HLF cells, and inhibited HepG2  colony formation. While, relative to untreated control cells, GB1101 did not influence the proliferation of HepG2 cells, it significantly suppressed the invasion of HLF cells and significantly inhibited their cell sphere formation.

On the basis of these data the Authors suggest that GB1101 may represent a novel drug candidate for the treatment of liver cancer.

The study is of interest and focuses on a topic that attracts great attention related to the modulation of specific modulation of TGF-beta in cancer.

I have a few comments on the paper on its present form.

  1. An in vivo study would have clearly strenghtened the study . Any chance for this, even preliminary data ?
  2. Any in vitro study availaible on the potential interference (synergistic, antagonistic) of   GB1101 with standard chemotherapeutic drugs ?
  3. The Authors should mention in the discussion section that TGF-beta is implicated in several disorders exerting beneficial or detrimental roles. In particular, this cytokine seems protective in autoimmune diseases (1) , plays a pathogenetic role in fibrosis (2) and exerts a complex role in certain pshychiatric disorders such as major depression (3). Hence, the Authors should indicate that GB1001 might have additional therapeutic role in fibrotic diseases but should be used with caution in autoimmune diseases and certain pshychiatric disorders  

1. Nicoletti F, et al. Blood levels of transforming growth factor-beta 1 (TGF-beta1) are elevated in both relapsing remitting and chronic progressive multiple sclerosis (MS) patients and are further augmented by treatment with interferon-beta 1b (IFN-beta1b).Clin Exp Immunol. 1998

2.  Fagone P,  et al., Emerging therapeutic targets for the treatment of hepatic fibrosis.Drug Discov Today. 2016 Feb;21(2):369-75. doi: 10.1016/j.drudis.2015.10.015. Epub 2015 Oct 30.

3. Petralia MC et al. The cytokine network in the pathogenesis of major depressive disorder. Close to translation?

4. Is there any data on the effect on TGF-beta production and toxicity of this compound in primary cells (e.g. lymphocytes)? It is important to know wheter at the concentrations found effective in these cancer cell lines the compound is toxic for primary cells. A concentration curve to evaluate cytotoxicity in primary cells is important

Author Response

Reviewer 3

This in vitro study demonstrates that synthetic pyrrole-imidazole (PI) polyamide GB1101 dose-dependently decreased TGF-β1 mRNA levels in HepG2 and HLF cells, and inhibited HepG2 colony formation. While, relative to untreated control cells, GB1101 did not influence the proliferation of HepG2 cells, it significantly suppressed the invasion of HLF cells and significantly inhibited their cell sphere formation.

On the basis of these data the Authors suggest that GB1101 may represent a novel drug candidate for the treatment of liver cancer.

The study is of interest and focuses on a topic that attracts great attention related to the modulation of specific modulation of TGF-beta in cancer.

I have a few comments on the paper on its present form.

An in vivo study would have clearly strenghtened the study . Any chance for this, even preliminary data ?

As indicated by the reviewer, it is necessary to validate the GB1101 effects on cancer by in vivo studies. We’ve not done yet but we previously reported the GB1101 effects on several types of diseases using rat or marmoset models. Therefore, we referred to a in vivo study for cancer treatment by GB1101 as one of perspectives in Discussion.

“There are several limitations in this study. First, we did not confirm inhibition of cancer progression via TGF-β1 suppression by GB1101 in protein level. However, we previously reported the GB1101 effects on various types of diseases such as nephropathy [20], hypertrophic scar [15], restenosis of arteries after injury [22], peritoneal sclerosis [23], and liver fibrosis [24] using rat or marmoset models. Therefore, the perspectives of this study is to evaluate whether GB1101 would actually be available to cancer therapy by in vivo study.”

(line 194–199)

Any in vitro study availaible on the potential interference (synergistic, antagonistic) of   GB1101 with standard chemotherapeutic drugs ?

As indicated by the reviewer, we should have investigated the potential interference of GB1101 with other chemotherapeutic drugs. We also described it as one of perspectives in Discussion.

“Second, given that polypharmacy using anti-cancer drugs and molecular drugs is a mainstream for cancer treatment, synergistic or antagonistic potential interference with GB1101 should have been investigated. Then we’re now planning to identify the drugs that enhance the GB1101 ability to kill the cancer cells.”

(line 199–202)

The Authors should mention in the discussion section that TGF-beta is implicated in several disorders exerting beneficial or detrimental roles. In particular, this cytokine seems protective in autoimmune diseases (1), plays a pathogenetic role in fibrosis (2) and exerts a complex role in certain pshychiatric disorders such as major depression (3). Hence, the Authors should indicate that GB1001 might have additional therapeutic role in fibrotic diseases but should be used with caution in autoimmune diseases and certain pshychiatric disorders

  1. Nicoletti F, et al. Blood levels of transforming growth factor-beta 1 (TGF-beta1) are elevated in both relapsing remitting and chronic progressive multiple sclerosis (MS) patients and are further augmented by treatment with interferon-beta 1b (IFN-beta1b).Clin Exp Immunol. 1998
  2. Fagone P, et al., Emerging therapeutic targets for the treatment of hepatic fibrosis.Drug Discov Today. 2016 Feb;21(2):369-75. doi: 10.1016/j.drudis.2015.10.015. Epub 2015 Oct 30.
  3. Petralia MC et al. The cytokine network in the pathogenesis of major depressive disorder. Close to translation?

As indicated by the reviewer, we described the beneficial or detrimental roles of TGF-β1, and discussed the accompanied benefits and side effects by GB1101 for liver cancer patients.

“In addition, pyridine derivative that inhibits TGF-β production improve HCV-related fibrosis and inflammation [25]. Therefore GB1101 administered for liver cancer patients may improve the coexisting liver fibrosis. Given that a cell-viability assay on cultured hepatocyte cell lines showed that PI polyamide treatments induced no significant reduction in viability or increase in cell death [26], there was no effect on cytotoxicity of PI polyamide on primary cells such as lymphocytes.

On the other hand, due to the multiple anti-inflammatory activities, TGF-β1 is thought to inhibit several aspects of the immune response, and this cytokine could have an immune-down-regulatory role in the pathophysiology of chronic progressive multiple sclerosis

[27]. Furthermore, it has been reported that TGF-β1 blood level of patients with major depressive disorder was lower compared to healthy controls [28]. Therefore, GB1101 has possibility to exacerbate the symptoms of such patients as side effects.”

(line 158−168)

  1. Is there any data on the effect on TGF-beta production and toxicity of this compound in primary cells (e.g. lymphocytes)? It is important to know wheter at the concentrations found effective in these cancer cell lines the compound is toxic for primary cells. A concentration curve to evaluate cytotoxicity in primary cells is important

Concerning the cytotoxicity of PI polyamides on cells, we previously performed a cell-viability assay on cultured hepatocyte cell lines and found that PI polyamide treatments induced no significant reduction in viability or increase in cell death.

“Given that a cell-viability assay on cultured hepatocyte cell lines showed that PI polyamide treatments induced no significant reduction in viability or increase in cell death [26], there was no effect on cytotoxicity of PI polyamide on primary cells such as lymphocytes.”

(line 160–162)

Round 2

Reviewer 2 Report

N/A

Reviewer 3 Report

The Authors have adequately addressed my criticisms